Improved genome of Agrobacterium radiobacter type strain provides new taxonomic insight into Agrobacterium genomospecies 4

http://orcid.org/0000-0001-7987-738X Gan Han Ming 1 2 3
Lee Melvin V.L. 3
http://orcid.org/0000-0003-1328-2978 Savka Michael A. 4 massbi@rit.edu
1 Deakin Genomics Centre, Deakin University , Geelong, VIC , Australia
2 Centre for Integrative Ecology, School of Life and Environmental Sciences, Deakin University , Geelong, VIC , Australia
3 School of Science, Monash University Malaysia , Petaling Jaya, Selangor , Malaysia
4 College of Science, The Thomas H. Gosnell School of Life Sciences, Rochester Institute of Technology , Rochester, NY , USA
Gelfand Mikhail
Electronic publication date: 2019 Feb 8
Publication date: 2019
Volume: 7
Electronic Location ID: e6366
Received 2018 Oct 23; Accepted 2018 Dec 20
Copyright: © 2019 Gan et al.
Copyright year: 2019
Copyright holder: Gan et al.
License: This is an open access article distributed under the terms of the Creative Commons Attribution License, which permits unrestricted use, distribution, reproduction and adaptation in any medium and for any purpose provided that it is properly attributed. For attribution, the original author(s), title, publication source (PeerJ) and either DOI or URL of the article must be cited.
License URL: https://creativecommons.org/licenses/by/4.0/

Keywords: Type strain, Average nucleotide identity, Phylogenomics, Agrobacterium radiobacter, Agrobacterium tumefaciens, Lipopolysaccharide, Agrobacterium, Ti plasmid

Funding: College of Science School of Life Sciences at Rochester Institute of Technology Michael A. Savka and Han Ming Gan received support from the College of Science and the Thomas H. Gosnell School of Life Sciences at Rochester Institute of Technology. The funders had no role in study design, data collection and analysis, decision to publish, or preparation of the manuscript.

==============================
The reported Agrobacterium radiobacter DSM 30174T genome is highly fragmented, hindering robust comparative genomics and genome-based taxonomic analysis. We re-sequenced the Agrobacterium radiobacter type strain, generating a dramatically improved genome with high contiguity. In addition, we sequenced the genome of Agrobacterium tumefaciens B6T, enabling for the first time, a proper comparative genomics of these contentious Agrobacterium species. We provide concrete evidence that the previously reported Agrobacterium radiobacter type strain genome (Accession Number: ASXY01) is contaminated which explains its abnormally large genome size and fragmented assembly. We propose that Agrobacterium tumefaciens be reclassified as Agrobacterium radiobacter subsp. tumefaciens and that Agrobacterium radiobacter retains it species status with the proposed name of Agrobacterium radiobacter subsp. radiobacter. This proposal is based, first on the high pairwise genome-scale average nucleotide identity supporting the amalgamation of both Agrobacterium radiobacter and Agrobacterium tumefaciens into a single species. Second, maximum likelihood tree construction based on the concatenated alignment of shared genes (core genes) among related strains indicates that Agrobacterium radiobacter NCPPB3001 is sufficiently divergent from Agrobacterium tumefaciens to propose two independent sub-clades. Third, Agrobacterium tumefaciens demonstrates the genomic potential to synthesize the L configuration of fucose in its lipid polysaccharide, fostering its ability to colonize plant cells more effectively than Agrobacterium radiobacter.

Introduction

The taxonomy and phylogeny of the genus Agrobacterium has proven to be complex and controversial. Bacteria of the genus Agrobacterium have been grouped into six species based on the disease phenotype associated, in part, with the resident disease-inducing plasmid. Among those six species are Agrobacterium tumefaciens causing crown gall on dicotyledonous plants, stone fruit and nut trees and Agrobacterium radiobacter that is not known to cause plant diseases of any kind (Bouzar & Jones, 2001; Conn, 1942; Kerr & Panagopoulos, 1977; Panagopoulos, Psallidas & Alivizatos, 1978; Riker et al., 1930; Starr & Weiss, 1943; Süle, 1978). An alternative classification approach grouped Agrobacterium organisms into three biovars based on physiological and biochemical properties without consideration of disease phenotype (Keane, Kerr & New, 1970; Kerr & Panagopoulos, 1977; Panagopoulos, Psallidas & Alivizatos, 1978). The species and biovar classification schemes do not coincide well, in a large part, because of the disease-inducing plasmids, tumor-inducing (pTi) and hairy root-inducing (pRi), are readily transmissible plasmids (Young et al., 2001).

Many widely used approaches for bacterial species definition include composition of peptidoglycan, base composition of DNA, fatty acid and 16S rDNA sequence (Stackebrandt et al., 2002) in addition to newer methods based on the whole-genome analysis (Coutinho et al., 2016; Jain et al., 2018), horizontal gene transfer analysis (Bobay & Ochman, 2017) or the core genome analysis (Moldovan & Gelfand, 2018) which is used in the present study. The genus Agrobacterium is a prime example with many proposals and oppositions regarding the amalgamation of Agrobacterium and Rhizobium over the last three or four decades (Farrand, Van Berkum & Oger, 2003; Gaunt et al., 2001; Young et al., 2001, 2003). However, more recent studies appear to favor the preservation of the genus Agrobacterium backed by strong genetic and genomic evidence (Gan & Savka, 2018; Ramírez-Bahena et al., 2014). Within the genus Agrobacterium, the taxonomic status of Agrobacterium radiobacter and Agrobacterium tumefaciens remains contentious (Sawada et al., 1993; Young, 2008; Young, Pennycook & Watson, 2006). Agrobacterium radiobacter (originally proposed as Bacillus radiobacter) is a non-pathogenic soil bacterium associated with nitrogen utilization isolated more than a century ago in 1902 (Beijerinck & Van Delden, 1902; Conn, 1942). On the other hand, Agrobacterium tumefaciens (previously Bacterium tumefaciens) is a plant pathogen capable of inducing tumorigenesis (Smith & Townsend, 1907). However, the descriptive assignment for Agrobacterium tumefaciens was later found to be contributed by a set of genes located on the large Ti plasmid that can be lost (Gordon & Christie, 2014). In other words, the curing of Ti plasmid in Agrobacterium tumefaciens will change its identity to the non-pathogenic species, Agrobacterium radiobacter. Furthermore, comparative molecular analysis based on single-copy housekeeping genes also supports the close relatedness of Agrobacterium radiobacter and Agrobacterium tumefaciens, blurring the taxonomic boundaries between these species (Mousavi et al., 2015; Shams et al., 2013). As taxa are reclassified into different populations that do not conform to the characteristics of the original description, the given names lose their significant and descriptive importance. Consistent with the Judicial Commission according to the Rules of the International Code of Nomenclature of Bacteria, Tindall (2014) concluded that the combination of Agrobacterium radiobacter has priority over the combination Agrobacterium tumefaciens when the two are treated as members of the same species since Agrobacterium radiobacter was the first proposed and described in 1902 whereas Agrobacterium tumefaciens was first proposed and described in 1907) (Tindall, 2014). However, given that Agrobacterium tumefaciens has been more widely studied than Agrobacterium radiobacter due to its strong relevance to agriculture (Bourras, Rouxel & Meyer, 2015), it remains unclear but interesting to see if the broader scientific community will obey this rule by adopting the recommended species name change in future studies.

To our knowledge, a detailed comparative genomics analysis of Agrobacterium radiobacter and Agrobacterium tumefaciens type strains has not been reported despite their genome availability (Zhang et al., 2014). The high genomic relatedness of both type strains was briefly mentioned by Kim & Gan (2017) through whole genome alignment and pairwise nucleotide identity calculation from homologous regions. However, evidence is now mounting that the Agrobacterium radiobacter DSM 30147T reported by Zhang et al. (2014) is contaminated, warranting immediate investigation (Jeong, Pan & Park, 2016). The assembled genome is nearly 7 megabases, the largest among Agrobacterium currently sequenced at that time with up to 6,853 predicted protein-coding genes contained in over 600 contigs. At sequencing depth of nearly 200×, its genome assembly is unusually fragmented even for a challenging microbial genome (Utturkar et al., 2017). Furthermore, the phylogenomic placement of Agrobacterium radiobacter DSM 30147T based on this genome assembly has been questionable as evidenced by its basal position and substantially longer branch length relative to other members of the species (Gan & Savka, 2018). The overly fragmented nature of this assembly also precludes fruitful comparative genomics focusing on gene synteny analysis. More importantly, analysis done on a contaminated assembly but with the assumption that it is not, will likely lead to incorrect biological interpretations (Allnutt et al., 2018).

In this study, we sequenced the whole genome of Agrobacterium radiobacter using a type strain that was sourced from the National Collection of Plant Pathogenic Bacteria (NCPPB). We produced a contiguous genome assembly exhibiting genomic statistics that are more similar to other assembled Agrobacterium genomes. We show here, through comparative genomics and phylogenetics, that the previously assembled Agrobacterium radiobacter DSM 30147T genome contains substantial genomic representation from another Agrobacterium sp. isolated and sequenced by the same lab, consistent with our initial suspicion of strain contamination. Using the newly assembled genome for subsequent comparative analysis, we provide genomic evidence that Agrobacterium radiobacter DSM 30147T and Agrobacterium tumefaciens B6T are the same species. However, strain DSM 30147T should not be considered as a merely non-tumorigenic strain of Agrobacterium tumefaciens as substantial genomic variation exists between these two type strains notably in the nucleotide sugar metabolism pathway that may contribute to their ecological niche differentiation.

Materials and Methods

DNA extraction and whole genome sequencing

Approximately 10 bacterial colonies were scrapped using a sterile P200 pipette tip from a 3-day-old nutrient agar culture and resuspended in lysis buffer with proteinase K (Sokolov, 2000) followed by incubation at 56 °C for 3 h. DNA purification was performed as previously described. The extracted DNA was normalized to 0.2 ng/μL and prepared using the Nextera XT library preparation kit (Illumina, San Diego, CA, USA) according to the manufacturer’s instructions. The library was sequenced on an Illumina MiSeq desktop sequencer located at the Monash University Malaysia Genomics Facility (2 × 250 bp run configuration) that routinely sequences mostly decapod crustacean mitogenomes (Gan, Tan & Austin, 2016a; Gan et al., 2016b; Tan et al., 2015) and occasionally microbial genomes (Gan et al., 2014, 2015; Wong et al., 2014) without prior history of processing any member from the Agrobacterium genomospecies 4.

De novo assembly and genome completeness assessment

Raw paired-end reads were adapter-trimmed using Trimmomatic v0.36 (Bolger, Lohse & Usadel, 2014) followed by error-correction and de novo assembly using Spades Assembler v3.9 (Bankevich et al., 2012) (See Data S1 for specific trimming and assembly settings). Genome completeness was assessed with BUSCOv3 (Rhizobiales database) (Waterhouse et al., 2017).

Protein clustering

Gene prediction used Prodigal v2.6 (Hyatt et al., 2010). Clustering of the predicted coding sequence was performed with CD-HIT-EST using the settings “-C 0.95, -T 0.8” (Li & Godzik, 2006). Identification of unique and shared clusters were done using basic unix commands, for example, csplit, grep, sort and uniq. The specific commands used and files generated during clustering can be found in the Zenodo database (https://doi.org/10.5281/zenodo.1489356).

Phylogenetic analysis

Reconstruction of the Agrobacterium phylogeny used PhyloPhlAN (Segata et al., 2013). PhyloPhlAN is a bioinformatic pipeline that identifies conserved proteins (400 markers) from microbial genomes and uses them to construct a high-resolution phylogeny using maximum likelihood inference approach (Price, Dehal & Arkin, 2010). For single gene tree construction, protein sequences were aligned with mafft v7.3 (Katoh & Standley, 2013) using the the most accurate setting (–localpair –maxiterate 1000) followed by phylogenetic tree construction via IqTree v1.65 with optimized model (Kalyaanamoorthy et al., 2017; Nguyen et al., 2014). Visualization and annotation of phylogenetic trees was performed with Figtree v1.4.3 (http://tree.bio.ed.ac.uk/software/figtree/).

Pan-genome construction and phylogenomics

Whole genome sequences were reannotated with Prokka v1.1 using the default setting (Seemann, 2014). The Prokka-generated gff files were used as the input for Roary v3.12.0 to calculate the pan-genome (Page et al., 2015). Maximum likelihood tree construction of the core-genome alignment and tree visualization used FastTree2 v2.1.10 (-nt -gtr) (Price, Dehal & Arkin, 2010) and FigTree v 1.4.3, respectively. Input and output files associated with the Roary analysis have been deposited in the Zenodo database (https://doi.org/10.5281/zenodo.1489356).

Detection and visualization of Ti plasmid

Genome sequences of each member of the genomospecies 4 except for the problematic DSM 37014T strain were used as the query for blastN search (e-value 1e−100) against the octopine-type Ti plasmid (Altschul et al., 1990). The result of the similarity search was subsequently visualized in Blast Ring Image Generator v0.95 (Alikhan et al., 2011).

Genome annotation and KEGG pathway reconstruction

Whole genome sequences of Agrobacterium tumefaciens B6T and Agrobacterium radiobacter NCPPB 3001T were submitted to the online server GhostKoala (Kanehisa, Sato & Morishima, 2016b) for annotation and the annotated genomes were subsequently used to reconstruct KEGG pathways (Kanehisa et al., 2016a) in the same webserver. Identification of proteins with TIGRFAM signatures of interest (Haft, Selengut & White, 2003) used HMMsearch v3.1b2 with the option “–cut_tc” activated to filter for only protein hits passing the TIGRFAM trusted cutoff values (Johnson, Eddy & Portugaly, 2010).

Results

An improved Agrobacterium radiobacter type strain genome

Raw sequencing data and whole genome assembly for strains B6 and NCPPB3001 reported in this study are linked to the NCBI Bioproject IDs PRJNA300485 and PRJNA300611, respectively. The newly assembled genome of Agrobacterium radiobacter type strain that was sourced from the NCPPB is approximately 30% smaller than the first reported Agrobacterium radiobacter DSM 30147T genome with 96% less contigs (22 vs 612), 20-fold longer N50 (480 vs 23 kb) and assembled length that is much more similar to other Agrobacterium spp. (Table 1). In addition, it is near-complete with 685 out of 686 BUSCO Rhizobiale single-copy genes detected as either partial or complete with minimal evidence of contamination as indicated by the near absence of duplicated single-copy gene(<0.1%). On the contrary, the current DSM 30147 genome is missing 25.1% of the single copy gene with up to 34.8% duplication rate. At the time of this manuscript writing, another genome of Agrobacterium radiobacter type strain that was sourced from another culture collection centre, for example, the Belgian Coordinated Collections of Microorganisms has been deposited in the NCBI wgs database (Agrobacterium radiobacter LMG140T; Table 1) with assembly statistics that are highly similar to the type strain genome reported in this study.

Table 1 Genome statistics of publicly available Agrobacterium genomospecies 4 whole genome sequences.

Assembly accession	Strain	Isolation source	Country	Size	GC%	# Contig	
GCF_900045375	B6	Apple Gall (Iowa)	USA	5.8	59.07	4	
GCF_001541315*	B6	Apple Gall (Iowa)	USA	5.6	59.32	52	
GCF_001692245	B140/95	Peach/Almond Rootstock	USA	5.7	59.23	45	
GCF_002179795	LMG 215	Humulus lupulus gall (USA)	USA	5.4	59.48	33	
GCF_000233975	CCNWGS0286	R. pseudoacacia nodules	China	5.2	59.53	49	
GCF_900011755	Kerr 14 = LMG 15 = CFBP 5761	Soil around Prunus dulcis	Australia	5.9	59.04	5	
GCF_002591665	186	English Walnut gall	California	5.7	59.42	22	
GCF_002008215	LMG 140 = NCPPB 3001 = CFBP 5522= DSM 30147	Saprobic soil	Germany	5.5	59.34	22	
GCF_000421945	LMG 140 = NCPPB 3001 = CFBP 5522 = DSM 30147	Saprobic soil	Germany	7.17	59.86	612	
GCF_001541305*	LMG 140 = NCPPB 3001 = CFBP 5522 = DSM 30147	Saprobic soil	Germany	5.5	59.36	22	
GCF_900012605	CFBP 5621	Lotus corniculata, root tissue commensal	France	5.4	59.32	3	
GCF_003031125	LAD9 (CGMCC No. 2962)	Landfill leachate treatment system	China	5.9	59.13	49	
GCF_000384555	224MFTsu31	Rhizosphere of L. luteus in Hungary, formerly R. lupini H13-3	USA	4.8	59.73	21	
GCF_900188475	719_389	Rhizosphere and endosphere of Arabidopsis thaliana.	USA	4.9	59.73	18	
GCF_000384555	UNC420CL41Cvi	Plant associated	USA	5	59.69	18	
Note:

* Reported in this study.

The inflated genome size of Agrobacterium radiobacter DSM 30147(T) is due to technical errors

Instead of sharing a recent common ancestor as would be expected for a recently duplicated gene, the duplicated single copy genes coding for seryl-tRNA synthetase in Agrobacterium radiobacter DSM 30147T were placed in two distinct clusters with one affiliated to genomospecies 4 and the other affiliated to genomospecies 7 (Fig. 1A). Such an unexpected clustering pattern raises the suspicion of genome assembly from two or more non-clonal bacterial strains. In addition, by performing comparison at the genome-scale based on whole proteome clustering of Agrobacterium radiobacter DSM 30147T/NCPPB 3001T (Previous study, GCF_000421945; This study, GCF_001541305), A. sp. TS43 (unpublished, GCF_001526605) and Agrobacterium tumefaciens B6 (GCF_001541315), we observed a high number of proteins that were exclusively shared between Zhang et al. Agrobacterium radiobacter DSM 30147 and A. sp. TS43 belonging to genomospecies 7 (Fig. 1B). Coincidentally, despite not sharing the same Bioproject ID, the whole genomes of strains DSM 30147T and TS43 were sequenced by the Zhang et al., and submitted to NCBI on the same date, 30 May 2013, hinting strain contamination during sample processing in the lab.

Figure 1 Phylogenetic and genomic evidence indicating contamination in the published A. radiobacter DSM 30147T genome.

(A) Maximum likelihood phylogenetic tree of seryl-tRNA synthetases from Agrobacterium genomospecies 4 and 7. Codes after the tildes are contigs containing the corresponding homologs. Node labels indicate ultra-fast bootstrap support value and branch length indicates number of substitutions per site. Duplicated homologs in the problematic A. radiobacter DSM 30147 genome were colored red. (B) Venn diagram of the core proteome of selected Agrobacterium strains from genomospecies 4. Numbers in the overlapping regions indicate the number of coding sequences (CDS) that shared by two or more groups at 95% nucleotide identity cutoff.

Genome-scale average nucleotide identity calculation supports the amalgamation of Agrobacterium radiobacter and Agrobacterium tumefaciens into a single genomospecies

Single gene tree shows that Agrobacterium radiobacter NCPPB 3001T and Agrobacterium tumefaciens B6T belong to the genomospecies 4 clade (Fig. 1A), corroborating with the PhyloPhlAN phylogenomic tree that was constructed based on the alignment of 400 universal single-copy proteins (Fig. S1). The pairwise average nucleotide identity (ANI) among strains within this clade is consistently more than 95% further supporting their affiliation to the same genomospecies (Fig. 2) (Coutinho et al., 2016; Jain et al., 2018). As expected, pairwise ANI of less than 92% was observed when they were compared with strains from genomospecies 7 (strains RV3 and Zutra 3/1). A 100% pairwise ANI was observed between Agrobacterium radiobacter type strains that were sourced from NCPPB and LMG. In addition, non-type strains B140/95 and CFBP5621 also exhibit a strikingly high pairwise ANI (>99%) to the type strains of Agrobacterium tumefaciens and Agrobacterium radiobacter, respectively, leading to the formation of sub-clusters within genomospecies 4 (Fig. 2).

Figure 2 A heatmap showing the hierarchical clustering of Agrobacterium strains based on genomic distance.

Values in boxes indicate pairwise average nucleotide identity. Horizontal colored bar below the heatmap indicate the genomospecies assigned to each genome (G7, genomospecies 7; G4, genomospecies 4). Boxed labels indicate genomes sequenced in this study.

Is Agrobacterium radiobacter a non-tumorigenic strain of Agrobacterium tumefaciens?

A majority of the currently sequenced strains from genomospecies 4 are non-tumorigenic as evidenced by the near complete lack of genomic region with significant nucleotide similarity to the octopine-type Ti reference plasmid (Fig. 3). Of the 14 genomes analyzed, only strains B6T and B140/95 exhibit a complete coverage of the Ti plasmid with near 100% sequence identity while strain 186 shows hits mainly to the essential gene clusters of a Ti plasmid such as the vir gene cluster (black rings and gene labels in Fig. 3) at a substantially lower sequence identity (50% < x < 90%) (Fig. 3), suggesting that it may be harboring a dissimilar variant of Ti plasmid, for example, different opine type. In addition, although lacking hits to the virulence gene of the Ti plasmid, the tra and trb clusters involved in plasmid conjugal transfer are present in strains Kerr 14, CCNWGS0286 and UNC420CL41Cvi. Despite belonging to the same genomospecies, core genome alignment and phylogenomic analysis indicates that Agrobacterium radiobacter NCPPB3001T is sufficiently divergent from Agrobacterium tumefaciens B6T leading to their separation into two distinct sub-clusters (Fig. 4A). This is also resonated by their different sub-cluster placement in the pairwise ANI heatplot (Fig. 2). Furthermore, strains from both subclades could be broadly differentiated by the set of core accessory genes that they harbor (Fig. 4B). Therefore, even though Agrobacterium radiobacter does not harbor a Ti plasmid, it cannot be considered as a non-tumorigenic strain of Agrobacterium tumefaciens given multiple lines of evidence indicating its substantial genomic divergence from Agrobacterium tumefaciens.

Figure 3 Prevalence and sequence conservation of the octopine-type Ti plasmid among Agrobacterium genomospecies 4.

Each genome (labelled 1–15) is represented by a colored ring shaded based on nucleotide percentage similarity to the reference Ti plasmid (min. 50%; max. 100%). The outermost ring highlights the gene regions involved in tumorigenesis (vir, iaa and ipt) and plasmid conjugation (trb and tra). Asterisks indicate genomes sequenced in this study.

Figure 4 Genomic divergence among genomospecies 4 strains.

(A) Unrooted maximum likelihood tree constructed based on the core genome alignment. Branch length and node labels indicate number of substitutions per site and FastTree2 SH-like support values, respectively. Putative subclades were colored blue, red and purple (B) Distribution of accessory (non-core) gene clusters among strains determined with Roary and plotted with the perl script roary2svg.pl (https://github.com/sanger-pathogens/Roary/blob/master/contrib/roary2svg/roary2svg.pl). A total of 7,906 accessory gene clusters were identified by Roary and the number of accessory genes presence in each genome are shown in the most right column. Vertical gray lines/bars along the plot indicate presence of accessory gene. Asterisks indicate genomes sequenced in this study.

Agrobacterium genomospecies 4 strains differ in their genomic potential for nucleotide sugar metabolism

Individual comparison of the reconstructed KEGG pathways in Agrobacterium tumefaciens (Fig. 5A) and Agrobacterium radiobacter (Fig. 5B) revealed stark contrast in the anabolism of dTDP-L-rhamnose which is commonly found in the O-antigen of lipopolysaccharide (LPS) in gram-negative bacteria. Surprisingly, the entire enzyme set required for the generation of dTDP-L-rhamnose from D-glucose-phosphate (Table 2) is absent in Agrobacterium tumefaciens B6, suggesting that this common nucleotide sugar may be absent from the LPS O-antigen of strain B6. A manual inspection of the accessory genes uniquely shared by Agrobacterium tumefaciens strains B6 and B140/95 identified a homolog cluster containing GDP-L-fucose synthase (EC 1.1.1.271) that is involved in the enzymatic production of GDP-L-fucose from GDP-4-dehydro-6-deoxy-D-mannose and NADH (Table 2; Fig. 5C). As expected, the genes coding for this enzyme and GDP-mannose 4,6-dehydratase involved in the conversion of GDP-alpha-D-mannose to GDP-4-dehydro-6-deoxy-D-mannose, are absent in the Agrobacterium radiobacter NCPPB3001 genome (Fig. 5D). Intriguingly, HMMsearch scan revealed the presence of two protein hits to the TIGR01479 HMM profile in Agrobacterium tumefaciens B6 that corresponds to D-mannose 1,6-phosphomutase (EC 5.4.2.8) required for the synthesis of D-mannose 6-phosphate. In addition to strain B6, its close relative, strain B140/95, and a more distantly related strain Kerr14 also harbor two copies of this gene. However, one of the D-mannose 1,6-phosphomutases in strain Kerr14 is more divergent with a lower TIGRFAM HMM sequence score (Table 2). Furthermore, it exhibits less than 70% protein identity to the Agrobacterium tumefaciens B6 and B140/95 homologs, forming a private protein cluster in the pan-genome (data not shown).

Figure 5 KEGG pathway of nucleotide sugar metabolism associated with Agrobacterium lipopolysaccharide synthesis.

(A & B) genomic potential of A. tumefaciens B6 and A. radiobacter DSM 30147, respectively, in the biosynthesis of dTDP-L-rhamnose. (C & D) genomic potential of A. tumefaciens B6 and A. radiobacter DSM 30147, respectively, in the biosynthesis of GDP-L-Fucose. Numbers in boxes indicate Enzyme Commission numbers. White and green boxes indicate absence and presence of the corresponding enzymes, respectively, based on GhostKoala annotation (Kanehisa, Sato & Morishima, 2016b).

Table 2 Identification of Agrobacterium proteins with TIGRFAM domains involved in the biosynthesis of nucleotide sugar.

Assembly ID	Strain	TIGR01479 (EC 5.4.2.8)	TIGR01472 (EC 4.2.1.47)	TIGR01207 (EC 2.7.7.24)	TIGR01181 (EC 4.2.1.46)	TIGR01221 (EC 5.1.3.13)	TIGR01214 (EC 1.1.1.133)	
1st hit	2nd hit	
GCF_900045375	B6	690.2	566.6	589.5					
GCF_001541315	B6	690.2	566.6	589.5					
GCF_001692245	B140/95	690.2	566.6	589.5					
GCF_900011755	Kerr14	691.3	690.2	428.6*					
GCF_001541305	NCPPB3001	690.2			494.6	488.5	215.4	331.5	
GCF_002008215	LMG140	690.2			494.6	488.5	215.4	331.5	
GCF_900012605	CFBP5621	689.3			494.6	489.5	215.4	331.5	
GCF_002591665	186	689.3			494.6	488.5	215.4	331.8	
GCF_003031125	LAD9	688.5			494.4	487.9	215.4	329.9	
GCF_000233975	CCNWGS	644.8			494.6	487.5	215.4	331.8	
GCF_002179795	LMG215	690.2							
GCF_000384555	224MFTsu31	644.8							
GCF_000482285	UNC420CL41Cvi	644.8							
GCF_900188475	719_389	687.5							
Notes:

Numbers indicate bit scores calculated based on protein alignment to the model with higher scores indicating stronger and more significant hits.

* Formed a separate protein cluster from the rest of genomospecies 4 GDP-mannose-4,6-dehydratase orthologs (<70% pairwise protein identity).

Discussion

We re-sequenced the genome of Agrobacterium radiobacter type strain using strain directly obtained from NCPPB. The assembled Agrobacterium radiobacter genome reported in this study exhibits assembly statistics that are consistent with a high-quality draft genome such as high genome completeness and contiguity, near-zero contamination/duplication and comparable genome size to other closely related strains (Gan, Lee & Savka, 2018; Parks et al., 2015). Furthermore, given the improved contiguity and dramatic reduction in the number of contigs of this newly assembled draft genome, we recommend using this genome in place of the previously published draft genome for future Agrobacterium comparative studies.

The distinct separation of Agrobacterium genomospecies 4 and 7 at 95% ANI cutoff corroborates with the previously established “genomic yardstick” for species differentiation (Konstantinidis & Tiedje, 2005; Richter & Rosselló-Móra, 2009). Using this percentage cutoff, the ANI approach has been successfully used to provide a near “black-and-white” pattern of species separation in even some of the most diverse bacterial genera such as Pseudomonas, Arcobacter and Stenotrophomonas (Pérez-Cataluña et al., 2018; Tran, Savka & Gan, 2017; Vinuesa, Ochoa-Sánchez & Contreras-Moreira, 2018). Given the increasing evidence highlighting the robustness and reliability of the ANI approach in species delineation, the pairwise ANI between Agrobacterium tumefaciens and Agrobacterium radiobacter type strains that is at least 2.5% higher than the 95% cutoff value is rigorous evidence that they belong to the same genomospecies, effectively serving as the final nail in the coffin for the decade-long debate on their taxonomic status. The amalgamation of Agrobacterium radiobacter and Agrobacterium tumefaciens into a single species have been repeatedly suggested in the past few years but was complicated by the special status of Agrobacterium tumefaciens as the type species of the genus Agrobacterium despite the priority that Agrobacterium radiobacter has over Agrobacterium tumefaciens as it was isolated and described 3 years before Agrobacterium tumefaciens (Young et al., 2001, 2003). Despite sharing numerous morphological and biochemical features, differences in genomic features such as pairwise ANI, phylogenomic clustering and core accessory gene contents do exist among members in Agrobacterium genomospecies 4 that can facilitate the identification of genotypic and phenotypic variants to accurately delimit sub-species relationships in the future (Brenner, Staley & Krieg, 2000; Jezbera et al., 2011; Meier-Kolthoff et al., 2014; Tan et al., 2013).

To date the LPS for both type strains have been determined (De Castro et al., 2002, 2004). In stark contrast to Agrobacterium radiobacter, the Agrobacterium tumefaciens LPS consists of D-arabinose and L-fucose that have yet been reported to date in another members of the genus Agrobacterium (De Castro et al., 2002). The presence of the L configuration of fucose is considered to be rare even among plant pathogenic bacteria but may be associated with the ability of Agrobacterium tumefaciens to colonize or bind to wounded plant cell (Lippincott, Whatley & Lippincott, 1977; Whatley et al., 1976; Whatley & Spiess, 1977). It has been previously shown that the LPS of Agrobacterium tumefaciens but not Agrobacterium radiobacter can bind to the plant cells thus providing protection against subsequent infection by pathogenic strains (Whatley et al., 1976). The presence and absence of nucleotide sugars in the O-chain constituent of LPS in both type strains corroborates with their observed genomic potential in the nucleotide sugar metabolism pathway thus underscoring the utility of comparative genomics in facilitating the prediction of microbial host range and ecological niche (Klosterman et al., 2011). For example, the absence of L-rhamnose and L-fucose in the LPS of Agrobacterium tumefaciens B6 and Agrobacterium radiobacter DSM30147, respectively, is consistent with the lack of genes coding for enzymes involved with the particular nucleotide sugar metabolism. Generation of Agrobacterium tumefaciens B6 LPS mutant via targeted gene deletion (Kaczmarczyk, Vorholt & Francez-Charlot, 2012) or the classical but more laborious transposon mutagenesis approach followed by characterization of the LPS mutant host-range and phytopathogenicity will be instructive (Gan et al., 2011; Reuhs et al., 2005).

Our current genomic sampling indicates that the Ti plasmid appears to be restricted to the Agrobacterium tumefaciens subclade. The maintenance of the Ti plasmid is metabolically taxing given its large size (Barker et al., 1983; Glick, 1995). Even if the Ti plasmid was conjugally transfer, for example, to Agrobacterium radiobacter, the inability of Agrobacterium radiobacter to colonize plant host as evidenced by its LPS incompatibility will not confer an advantage to the new plasmid host in a natural environment (Thomashow et al., 1980). Furthermore, in the absence of high density Acyl-homoserine lactone (AHL) signals which is required to trigger Ti plasmid conjugation (Fuqua & Winans, 1994; Pappas, 2008; Zhang, Wang & Zhang, 2002), the newly acquired Ti plasmid in Agrobacterium radiobacter may be cured in its natural soil habitat after a few generations. Although the spontaneous transfer of the Ti plasmid from tumorigenic Agrobacterium tumefaciens to Agrobacterium radiobacter K84 has been reported previously, strain K84 was re-classified based on a recent core gene analysis to Rhizobium rhizogenes K84 (Velázquez et al., 2010; Vicedo et al., 1996), reiterating the pervasive taxonomic inconsistency within the genus Agrobacterium that may have confound previous biological interpretations (De Ley et al., 1966; Lindström et al., 1995; Young, 2008). Given that a large majority of Agrobacterium genetics was performed during the pre-NGS era (Gan & Savka, 2018), it remains unknown as to how many Agrobacterium tumefaciens and Agrobacterium radiobacter strains have been molecularly misclassified due to their high genomic relatedness.

The inability to accurately identify plasmid and chromosomal-derived contigs among the draft genomes means that some of the core accessory genes among tumorigenic strains may be plasmid-derived and should be treated with caution as the low-copy-number Ti-plasmid is prone to curing in the absence of AHL signals. Despite the value of complete genome assembly in enabling the accurate partitioning of plasmid and chromosomal genomic region (Arredondo-Alonso et al., 2017), the representation of complete Agrobacterium genomes in current database is still very low as a majority of the genomes were assembled from short Illumina reads that cannot effectively span repetitive region (Wibberg et al., 2011; Wood et al., 2001). Furthermore, most Agrobacterium strains harbor multiple large plasmids that further complicate short-read-only assembly graph (Kado & Liu, 1981; Lowe et al., 2009; Shao et al., 2018). Given the currently available genomic resources for Agrobacterium, defining subspecies within the Agrobacterium genomospecies 4 based on the identification of lineage-specific gene set (Moldovan & Gelfand, 2018) will be challenging. However, we anticipate that the advent of high throughput long-read sequencing that can span large repetitive region in recent years is likely going to overcome this limitation allowing a more accurate depiction of microbial pangenome (Gan et al., 2012; Gan, Lee & Austin, 2017; Schmid et al., 2018a, 2018b). Future hybrid genome assemblies (Illumina and Nanopore/PacBio reads) of members from genomospecies 4 with comprehensive metadata and reliable phenotypic information, will be instructive.

Conclusions

Despite belonging to the same genomospecies, Agrobacterium tumefaciens and Agrobacterium radiobacter are by no means clonal at the chromosomal level and instead demonstrate sufficient genomic characters that qualify their separation into two sub-species. In addition, the difference in the LPS profile among two type strains will have implications to host specificity leading to geographical separation. In the spirit of preserving the naming of both species but at the same time respecting the taxonomic jurisdiction for strain priority, we propose Agrobacterium tumefaciens to be reclassified as Agrobacterium radiobacter subsp. tumefaciens and for Agrobacterium radiobacter to retains its species status with the proposed name of Agrobacterium radiobacter subsp. radiobacter.

Supplemental Information

Supplemental Information 1 Supplementary Figure 1. Maximum likelihood phylogeny of the genus Agrobacterium inferred based on the concatenated alignment of 400 single copy conserved proteins.

The tree was rooted with members from the species Rhizobium rhizogenes (labeled as Agrobacterium rhizogenes) as the outgroup. Blue and red-colored clades belong to Agrobacterium genomospecies 4 and 7, respectively. Node labels indicate local SH-like support values. Branch lengths indicate the number of substitutions per site.

Click here for additional data file.

Supplemental Information 2 Commands used for adapter trimming and microbial genome assembly.

Click here for additional data file.

Additional Information and Declarations

Competing Interests

Author Contributions

DNA Deposition

Data Availability

The authors declare that they have no competing interests.

Han Ming Gan conceived and designed the experiments, performed the experiments, analyzed the data, contributed reagents/materials/analysis tools, prepared figures and/or tables, authored or reviewed drafts of the paper, approved the final draft.

Melvin V.L. Lee performed the experiments, analyzed the data, contributed reagents/materials/analysis tools, prepared figures and/or tables, approved the final draft.

Michael A. Savka conceived and designed the experiments, analyzed the data, authored or reviewed drafts of the paper, approved the final draft.

The following information was supplied regarding the deposition of DNA sequences:

Raw sequencing data and whole genome assembly for strains B6 and NCPPB3001 reported in this study are linked to the NCBI Bioproject IDs PRJNA300485 and PRJNA300611, respectively.

The following information was supplied regarding data availability:

LMVK00000000.1, ASM154131v1: https://www.ncbi.nlm.nih.gov/assembly/GCF_001541315.1;

LMVJ00000000.1, ASM154130v1: https://www.ncbi.nlm.nih.gov/assembly/GCF_001541305.1;

Code and data are available at Han Ming Gan. (2018). Dataset for “Improved genome of Agrobacterium radiobacter type strain provides new taxonomic insight into Agrobacterium genomospecies 4” [Data set]. Zenodo. DOI 10.5281/zenodo.1489356.

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
