# Peer review of "Improved genome of Agrobacterium radiobacter type strain provides new taxonomic insight into Agrobacterium genomospecies 4"

_PeerJ, doi:10.7717/peerj.6366_

## Round 0.1 · original submission · Major Revisions

While the reviewers accept the findings of the study, they have found a number of reporting and editorial issues that need to be addressed.

·

Basic reporting

There are some suggestions that could help to make the paper more readable:

Abstract
1. 22 –«Obtained from NCPPB» should be deleted here.
2. 31 – The authors should either define genomospecies or write just "species"
3. 32 – Please explain, what is meant by "core genome".

Introduction
1. 31: the authors should define «genomospecies»
2. 46-50 – is this information necessary for this article? Apart from A. tumefaciens and A. radiobacter these species are not mentioned in the manuscript anywhere else. The authors should substitute this part with: «Bacteria of the genus Agrobacterium have been grouped into six species based on the disease phenotype associated, in part, with the resident disease-inducing plasmid (references). Among those six species are A. tumefaciens causing crown gall on dicotyledonous plants, stone fruit and nut trees and A. radiobacter that is not known to cause plant diseases of any kind (Zhang et al., 2014)»
3. 58-59 instead of the first sentence the authors should at least mention here widely used classic approaches to bacterial species definition (see Stackebrandt et al., 2002) as well as novel methods based on the whole-genome analysis (Coutinho et al., 2016), horizontal gene transfer analysis (Bobay and Ochman, 2017) or the core genome analysis, which is used in the present study (Moldovan and Gelfand, 2018).
4. Line 80: not clear what is meant by «the principle of priority». Probably the authors should explain that and use indirect speech here.
5. 87-88: «has not been reported» probably?
6. 99: not clear what the '4' in «genomospecies 4» stands for. The authors should explain that.
7. 111: not clear by which criterion the evidence is conclusive. Please consider #3 and emphasize which criteria for the species definition are used.

Methods
1. 125-127: Here the authors should mention that no Agrobacterium strains have been sequenced in this laboratory previously.
2. 129-134: The article should be supplemented with the list of commands employed to construct the genome as well as with Trimmomatic read quality scores.
3. 135-142: Again, the commands or just a list of parameters should be provided as a supplementary material. Here the authors should just list the programmes and expand a little on the meaning of key parameters.
4. 141-142: «Visualisation and annotation of phylogenetic trees was performed with FastTree2 …»
5. 152-156: Please provide references for BLAST and for the image generator.
6. 158-164: The authors should provide a reference for KEGG here.

Results
1. 181: In the Table 1 there is no LM140, only LMG140, which one is it?
2. 207-214: The authors should include here a reference to a study that employed a 95% identity cutoff, e.g. Coutinho et al., 2016.
3. Also, In the notation to fig. 1b, there is a 95% protein identity cutoff, the authors should either explain this choice or use a 95% nucleotide identity cutoff and compare CDSs. Along with that, fig. 1B should be mentioned and commented on in the main text.
4. The authors should indicate on fig. 2 which strains belong to which species and genomospecies; strains mentioned in the text should be highlighted.
5. On figure 3 it should be indicated, which strains belong to which genomospecies.
6. On figure 4 there also should be clade annotations: which clade corresponds to which species.
7. Figure 4B should be explained better. It's not clear what do grey areas and numbers to the right mean.
8. 232: Figure 3 depicts an alignment to Ti plasmid; probably Figure 2 should be referenced here.
9. 245: Probably fig. 5A should be referenced here?
10. 248: fig. 5B?
11. Objects on figure 5 are too small. I’m afraid the authors have to re-draw it. In the notation a reference for GhostKoala should be included

Experimental design

no comment

Validity of the findings

no comment

Additional comments

The authors should include a link to the constructed draft genome

Reviewer 2 ·

Basic reporting

The manuscript entitled “Improved genome of Agrobacterium radiobacter type strain provides new taxonomic insight into Agrobacterium genomospecies 4” reports a comparative genomics analysis of Agrobacterium species including re-sequenced genome of Agrobacterium radiobacter with dramatically improved assembly quality and contiguity. Based on obtained results, authors proposed a reclassification of A. radiobacter/ A. tumefaciens taxonomy.
The topic of research seems interesting and basic reporting is good. The ‘background’ section is clear and sufficient. In the ‘Methods’ and ‘Results’ sections several details were missing that the reader might consider crucial for the logic of the article.
One more moment that appears to be very confusing is numerous formatting inaccuracies, such as incorrect links to figures and Supplements. As an example, Supplement Fig. 1 is mentioned in a text but the Supplement section is absent; at the same time, Fig. 1B is present in the article but not mentioned in the article body. The ‘Declaration’ section with the information regarding the data availability is missing. The list of citations went under the title “Acknowledgements”. I would strongly suggest a careful proofreading of the article.

Experimental design

L136-139 “Clustering of the predicted proteins was performed with CD-HIT Identification of unique and shared clusters were done using basic unix commands e.g. csplit, grep, sort and uniq.”
Constructed gene groups should be made available through an open source repository such as GitHub for the reproducibility of the study that I consider to be extremely important in context of the sensitivity of the results to parameters of orthologs clustering.

L206 “phylogenetic tree that was constructed based on the alignment 400 universal single-copy proteins”.
Does it mean that the constructed pan-genome contain exactly 400 universal single-copy genes or any additional filter was applied for the set of core genes?
As one of the main results of the article is based on phyletic pattern of accessory genes, brief description of constructed pangenome such as general statistics of core, accessory and paralogous gene groups in the ‘Results’ section also would be interesting for the reader.

L242. the term ‘core accessory genome’ should be explained. Does it mean ‘the set of genes that are common for a phylogroup but are not common for the species’? As traditional model of pangenome separates all genes on the core, accessory and unique parts, the term requires a definition.

Moreover, it looks like there are methods to achieve a better technical standard. In particular, the methodology of subspecies classification based on phyletic pattern of accesory genes was proposed in (Moldovan and Gelfand 2018).

Validity of the findings

The findings seem reasonable while the description of the methodology should be improved to allow enough support for statements made in the article.

---

## Round 0.2 · accepted · Accept

The reviewers are satisfied with the revised text.

# ·

Basic reporting

The text and content of the article are better now. Accepted.

Experimental design

Experiments conform to a good research standard. I thank the authors for making scripts and data publicly available

Validity of the findings

no comment

Reviewer 2 ·

Basic reporting

no comment

Experimental design

no comment

Validity of the findings

no comment

Additional comments

no comment